# Associations of general and central adiposity with hypertension and cardiovascular disease among South Asian populations: a systematic review and meta-analysis

Federica Re [1,2] Ayodipupo S Oguntade [1] Bastian Bohrmann,[1] Fiona Bragg [1,3] Jennifer L Carter [1,3]

¹Clinical Trial Service Unit and Epidemiological Studies Unit (CTSU), Nuffield Department of Population Health, University of Oxford, Oxford, UK
²Medical Sciences Division, University of Oxford, Oxford, UK
³MRC Population Health Research Unit, Nuffield Department of Population Health, University of Oxford, Oxford, UK

**Correspondence to**
Dr Jennifer L Carter;
jennifer.carter@ndph.ox.ac.uk

## ABSTRACT

**Background** The relevance of measures of general and central adiposity for cardiovascular disease (CVD) risks in populations of European descent is well established. However, it is less well characterised in South Asian populations, who characteristically manifest larger waist circumferences (WC) for equivalent body mass index (BMI). This systematic review and meta-analysis provide an overview of the literature on the association of different anthropometric measures with CVD risk among South Asians.

**Methodology** MEDLINE and Embase were searched from 1990 to the present for studies in South Asian populations investigating associations of two or more adiposity measures with CVD. Random-effects meta-analyses were conducted on the associations of BMI, WC and waist-to-hip ratio (WHR) with blood pressure, hypertension and CVD. Quality assessment was performed using the Newcastle-Ottawa scale.

**Results** Titles and abstracts were screened for 7327 studies, yielding 147 full-text reviews. The final sample (n=30) included 2 prospective, 5 case-control and 23 cross-sectional studies. Studies reported generally higher risks of hypertension and CVD at higher adiposity levels. The pooled mean difference in systolic blood pressure (SBP) per 5 kg/m² higher BMI was 3 mmHg (2.90 (95% CI 1.30 to 4.50)) and 6 mmHg (6.31 (95% CI 4.81 to 7.81) per 13 cm larger WC. The odds ratio (OR) of hypertension per 5 kg/m² higher BMI was 1.33 (95% CI 1.18 to 1.51), 1.45 (95% CI 1.05 to 1.98) per 13 cm larger WC and 1.22 (95% CI 1.04 to 1.41) per 0.1-unit larger WHR. Pooled risk of CVD for BMI-defined overweight versus healthy-weight was 1.65 (95% CI 1.55 to 1.75) and 1.48 (95% CI 1.21 to 1.80) and 2.51 (95% CI 0.94 to 6.69) for normal versus large WC and WHR, respectively. Study quality was average with significant heterogeneity.

**Conclusions** Measures of both general and central adiposity had similar, strong positive associations with the risk of CVD in South Asians. Larger prospective studies are required to clarify which measures of body composition are more informative for targeted CVD primary prevention in this population.

## STRENGTHS AND LIMITATIONS OF THIS STUDY

⇒ A broad literature search, robust duplicate and blinded screening, and a firm quality assessment represent major strengths of this study.
⇒ Sensitivity analyses excluding large studies contributing the most to the models in the meta-analyses showed that associations remained largely unchanged.
⇒ While the review aims to assess risk across South Asia, most studies were conducted in India, thus limiting generalisability of findings to the rest of the subcontinent.
⇒ Cross-sectional design of studies included in the review increases the possibility of reverse causality.

## INTRODUCTION

Being overweight or obese is one of the leading risk factors for premature mortality, estimated to account for up to 4.7 million deaths yearly.[1–3] The Global BMI Mortality Collaboration, a meta-analysis of 239 prospective studies, reported higher risk of cardiovascular disease (CVD) with higher body mass index (BMI) across all regions, showing that overweight individuals (BMI >25 kg/m²) experienced 49% higher risk per 5 kg/m² higher BMI than individuals whose weight was within the healthy range (BMI 18.5–24.9 kg/m²).[4] Further meta-analyses across Europe, North America and East Asia have reached similar conclusions.[5–7] The shape and strength of the association between BMI and CVD risk, however, has been found to differ across certain subgroups, with weaker associations among South Asians—though few studies have focused on these populations.[8 9]

A prospective cohort study of 0.5 million adults from the city of Chennai, India, found that blood pressure was strongly and positively

associated with coronary heart disease (CHD) mortality (risk ratio (RR) 1.70 (95% CI 1.60 to 1.80) per 20 mmHg higher usual systolic blood pressure (SBP), but that BMI was little related to CHD mortality, despite increased BMI being a strong determinant of increased SBP.[8] Moreover, the Asia Cohort Consortium observed positive associations between overweight BMI and CVD death, but the relation was substantially weaker and not statistically significant in South Asians compared with East Asians (South Asians: HR 1.03 [95% CI 0.93 to 1.15] vs East Asians: HR 1.09 (95% CI 1.03 to 1.15)).[10] The Global BMI Collaboration concluded similar findings, describing that, among South Asians, the risk of CVD per $5 \text{ kg/m}^2$ higher BMI in overweight individuals was 10% and not statistically significant (HR 1.10 (95% CI 0.83 to 1.46)). This was considerably weaker than the associations observed in European populations (HR 1.56 (95% CI 1.54 to 1.58)).[4]

There is some evidence suggesting that South Asians are at higher risk of diabetes and CVD compared with Western populations.[11] However, the underlying pathophysiology leading to ethnic variations in the prevalence of hypertension (HTN) and the risk of CVD is poorly understood, although differences in body fat composition and distribution may explain these discrepancies. As an anthropometric measure, BMI does not distinguish between central and peripheral adiposity, or between fat and lean mass. Some evidence suggests that centrally distributed visceral fat and ectopic fat are associated with cardiovascular outcomes, independently of BMI.[12 13] Furthermore, there are marked differences between central and general adiposity in their associations with fasting glucose, diabetes and blood pressure, which lie on the causal pathway for CVD.[12 14] Therefore, research needs to assess whether measures of central adiposity are more important markers of disease risk in South Asian populations.

There is mixed evidence on whether higher BMI is associated with higher risk of CVD among South Asians and on which anthropometric measures are more strongly related with CVD risk in these populations. Hence, the purpose of this review is to provide an overview of the literature on the shape and strength of the association of BMI and CVD, including HTN, among South Asians and to understand whether alternative anthropometric measures are better indicators of adiposity-related CVD risk in this geographical area compared with BMI. Results can be used to develop research which better characterises the importance of adipose tissue and its distribution in CVD risk. Over time, this can help develop targeted preventative interventions and minimise health disparities.

## METHODS
### Search strategy
This systematic review followed the Cochrane Collaboration methods and adhered to the PRISMA reporting recommendations.[15] A predetermined review protocol was registered in the PROSPERO database (CRD42022308682). MEDLINE and Embase were searched combining terms relating to central and general adiposity and cardiovascular outcomes. Further terms were included to select for the geographical area of interest, namely South Asia (Afghanistan, Bangladesh, Bhutan, India, Maldives, Nepal, Pakistan and Sri Lanka). The MeSH terms were chosen from the thesaurus used for indexing the subject headings. Full details of the search strategy for MEDLINE and Embase are shown in online supplemental data S1. The search was limited to studies conducted on adults (18+ years) and published in English between 1 January 1990 and 1 January 2023.

### Eligibility criteria
Eligible studies included at least one measure of general (BMI, $\text{kg/m}^2$) and one measure of central adiposity (waist circumference (WC), waist-to-hip ratio (WHR), waist-to-height ratio (WHtR)) as the exposure and at least one cardiovascular outcome: SBP (mmHg), diastolic blood pressure (DBP, mmHg), HTN (persistent blood pressure >140/90 mmHg), coronary artery disease, peripheral vascular disease or CVD incidence/mortality.

The search included cross-sectional, case–control and cohort studies. To limit potential publication bias from the inclusion of small studies with chance findings that reported stronger-than-average results, studies examining clinical endpoints (eg, CVD, HTN) were included in the review only if they included at least 50 events, while those examining continuous outcomes (eg, blood pressure) were included if they included at least 100 participants. Studies that solely performed correlational analyses, those conducted among participants with prevalent diseases, or those on South Asian populations living outside of South Asia were excluded.

### Study selection and data extraction
Studies were imported into Covidence, an online systematic review management platform, for abstract and full-text screening.[16] After removal of duplicates, titles and abstracts were reviewed independently by two reviewers (FR and ASO). Studies that did not meet the inclusion criteria were excluded. The full texts of the remaining papers were reviewed independently by the same reviewers. Disagreements were resolved by discussion and consensus between the two reviewers or, where necessary, by involving a third reviewer (JC). A full list of included studies is reported in online supplemental data S2 and S3. Associations were recorded for each study.

### Quality assessment
Quality of included studies was assessed using validated cross-sectional, case–control and cohort adaptations of the Newcastle-Ottawa scale.[17 18] Studies that fulfilled a criterion were awarded a point for that criterion, while no point was awarded if the criterion was not fulfilled. Cohort and case–control studies could be awarded a maximum of nine points, while cross-sectional studies

could be awarded a maximum of 10 points. Studies were considered of high quality if they met at least 7 out of 9e criteria for cohort and case–control studies, and 8 out of 10 criteria for cross-sectional studies. The quality of a study did not determine its inclusion in the systematic review or meta-analysis. Details of the quality assessment are available in online supplemental data S4a/S4b/S4c.

### Data synthesis and analysis

For the dose–response meta-analyses, summary RR (95% CIs) per 5 kg/m$^2$ higher BMI, 13 cm higher WC and 0.1-unit higher WHR were calculated using random effects models due to substantial heterogeneity of the included studies. BMI was presented as a 5 kg/m$^2$ change to allow comparability with other large-scale studies which have used the same units.[4 10] Associations with WC and WHR were compared with those of BMI by scaling the measures to the same SD unit change. Scaling factors were based on the mean and SD reported by Taing *et al* since this study had the largest sample size (n=7601) and included all three adiposity measures of interest.[19] The pooled BMI SD in the study by Taing *et al*[19] is 4.6 kg/m$^2$, so a change in 5 kg/m$^2$ represents a 1.087 SD change. Accordingly, central adiposity measures were then scaled to the same SD change as 5 kg/m$^2$ BMI, which corresponded to a 13 cm increase in WC and 0.1-unit change in WHR. Additional fixed-effects models were also calculated. Formulas for random and fixed effects models are included in online supplemental data S5. For each study, the risk estimates from the model including the greatest number of confounders, but not intermediate factors (eg, diabetes, left ventricular hypertrophy), were used. The average of the natural logarithm of the RRs was calculated.[20] In cases where studies provided RRs (95% CIs) per unit higher adiposity measure, these were scaled to the desired units by exponentiating the RR (95% CIs) to the power of desired units. When studies only reported RRs separately for different subgroups (eg, age, sex or ethnicity), these subgroup estimates were combined using a fixed-effects model to obtain an overall estimate. Each study was thus only represented once in each main meta-analysis.

Where studies reported estimates for categories of anthropometric measures, estimates were log-transformed and used to calculate study-specific slopes and 95% CIs across categories of anthropometric measures, to generate overall study-specific RRs.[21 22] Where studies only reported total cases and controls, total numbers were divided evenly across the categories.[23] The mean or median of each category of each anthropometric measure was assigned to the corresponding RRs. For studies that did not report the mean or median of the anthropometric measures, the midpoint of the range of such categories was used as the mean. When the lowest or highest category was open-ended, the width of the interval was assumed to be the same as that of the adjacent category.[23] A likelihood ratio test was used to test non-linearity by assessing the difference between the linear and non-linear models.

Heterogeneity between studies was determined using a Q-test, and I$^2$ statistics were used to denote the percentage of total variability due to between-study heterogeneity. I$^2$>70% indicated high heterogeneity. To assess the robustness of the overall estimates, sensitivity analyses were undertaken removing one study at a time to determine whether results were influenced by large studies or studies with extreme results. Publication bias and small study effects were examined by inspecting funnel plots for asymmetry and with Egger's test. Analyses were conducted using Stata/MP V.17.0 (StataCorp).

## RESULTS

The initial search, after duplicate removal, included 7327 studies. Of these, 30 were included in this review. Figure 1 shows the number of papers excluded at each stage of the review process. At the screening stage, most exclusions were of studies looking at a single anthropometric measure only or those taking place in geographical regions not included in the review. On full-text assessment, most exclusions (n=85) were of studies performing only correlation analyses between anthropometric measures and CVD incidence. The number of participants in included studies varied between 140 and 59 037.[24 25] There were 25 (83.3%) included studies in India[19 24–47] and three (10%) in Bangladesh.[48–50] The remaining two studies were conducted in Mauritian[51] and Pakistani[52] populations, respectively. The final analyses included two prospective cohort studies (one looking at HTN[26] and one looking at CVD mortality[51]), four case-control studies[44–47] (looking at CVD outcomes) and 24 cross-sectional studies[19 24 25 27–43 48–50 52] (looking at HTN outcomes). In terms of exposure variables, 29 studies included BMI, 27 studies used WC, 21 studies used WHR and ten studies additionally used other measures, including WHtR (n=6) and hip circumference (n=2). Overall, 18 studies looked at HTN,[24 26 28–38 40 41 48–50] with or without blood pressure, eight looked at CVD,[25 42–47 52] three looked at blood pressure alone[19 27 39] and one study looked at CVD mortality.[51]

### Associations between anthropometric indices and blood pressure

A total of four studies were included in the analysis of blood pressure (SBP/DBP). Of the four studies looking at the relationship between BMI and blood pressure, all studies concluded that higher BMI was related to higher blood pressure, with stronger associations with SBP than DBP. This was reflected in the meta-analysis, which showed that the pooled mean difference per 5 kg/m$^2$ higher BMI was 3 mm Hg (2.90 (95% CI 1.30 to 4.50)) for SBP (figure 2) and 2 mm Hg (2.28 (95% CI 0.55 to 4.01)) for DBP (online supplemental data S6). For WC, the pooled mean change in blood pressure per 13 cm larger WC was approximately 6 mm Hg (6.31 (95% CI 4.81 to 7.81)) for SBP (figure 2) and 5 mm Hg (5.18 (95% CI 3.18 to 7.18)) for DBP (online supplemental

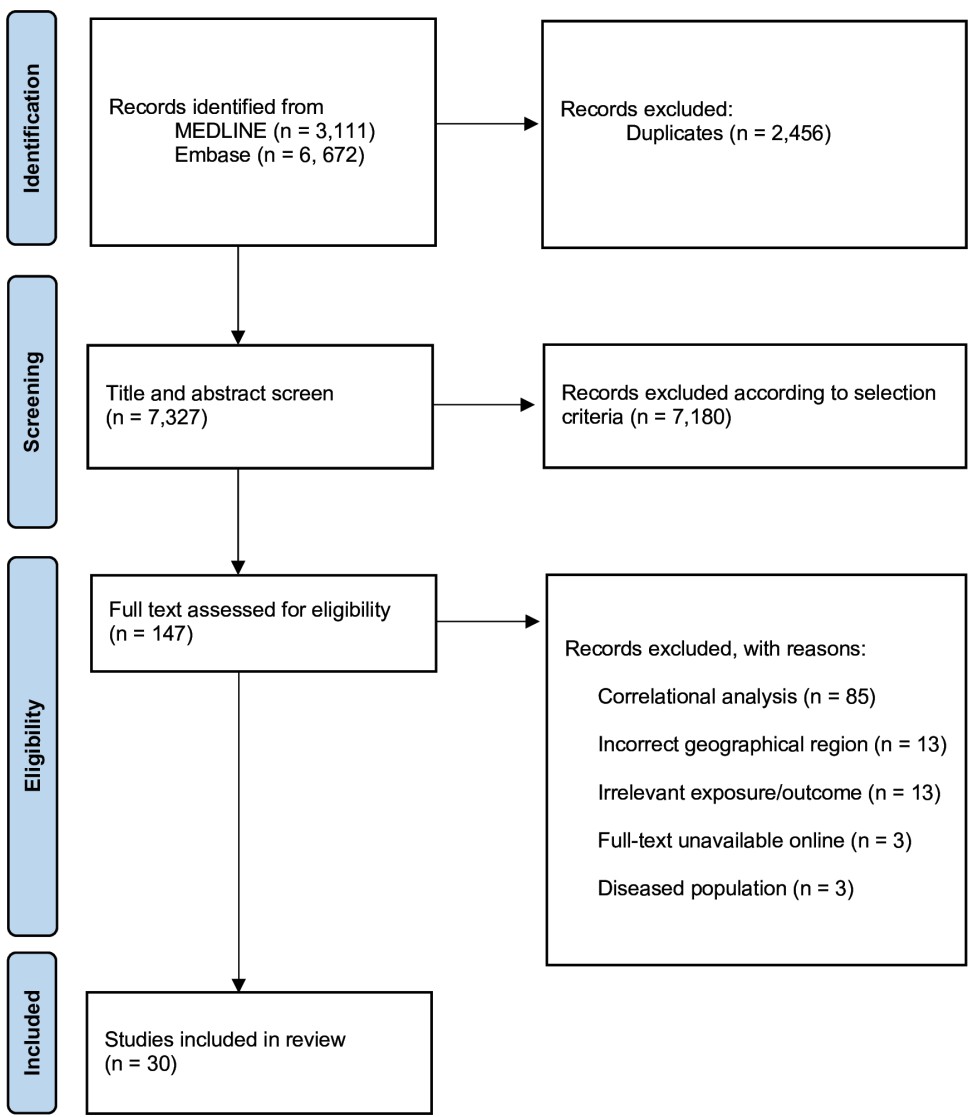

**Figure 1** Preferred Reporting Items for Systematic Reviews and Meta-Analyses (PRISMA) flow chart describing the systematic literature search and study selection.

data S7). Associations were not statistically significant per 0.1-unit change in WHR (SBP: 2.27 (95% CI −1.62–6.16); DBP: 1.94 (95% CI −1.42–5.29)). Findings of fixed-effects models are reported in online supplemental data S6 and S7.

### Associations between anthropometric indices and HTN

All studies looking at HTN risk reported positive associations between measures of general adiposity (BMI) and/or central adiposity (WC, WHR) with the risk of HTN. Five cross-sectional studies concluded that the risk of HTN was higher with a high BMI ($\geq 25\,kg/m^2$) compared with a large WC ($\geq 80\,cm$ in females, $\geq 90$ in males) or WHR ($\geq 0.8$ in females, $\geq 1.0$ in males). Eleven cross-sectional studies reported that measures of central adiposity, compared with BMI, showed stronger associations with HTN. Of these, seven reported stronger associations with WC[29 33 36 41 48–50] and four with WHR.[28 37 38 40] The pooled OR of HTN per $5\,kg/m^2$ higher BMI was 1.33 (95% CI 1.18 to 1.51; figure 3A). It was stronger for a 13 cm larger

WC (OR 1.45 (95% CI 1.05 to 1.98); figure 3B), but weaker for a 0.1-unit larger WHR (OR 1.22 (95% CI 1.05 to 1.41); figure 3C), though all associations remained statistically significant. Heterogeneity ($I^2$) was >99% in all models. Fixed-effects models showed weaker associations (online supplemental data S8).

### Association between anthropometric indices and CVD

The analyses of non-fatal and fatal CVD included one cohort, four cross-sectional and four case–control studies. Four studies concluded that there was a stronger association of high BMI than of high WC and WHR with risk of CVD and CVD mortality.[25 42 43 47] The remaining five studies concluded that measures of central adiposity, namely WC and/or WHR, showed stronger associations with CVD and CVD mortality than BMI.[44–46 51 52] The meta-analysis included 30 516 cases for the association of BMI with CVD, 31 274 cases for that of WC with CVD and 30 537 for that of WHR with CVD. The pooled risk of CVD for overweight versus normal-weight individuals, as determined by BMI,

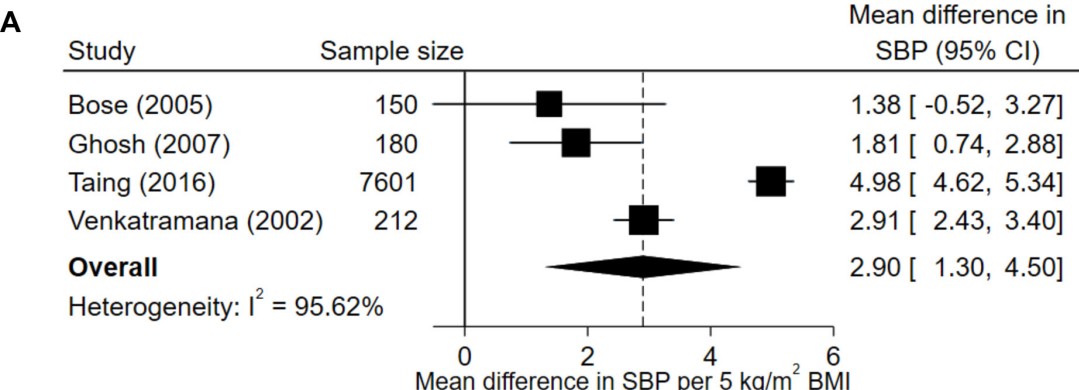

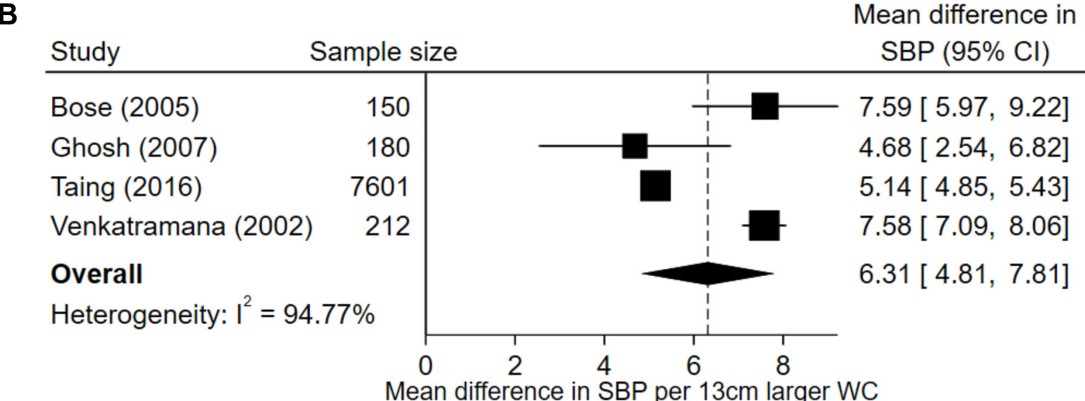

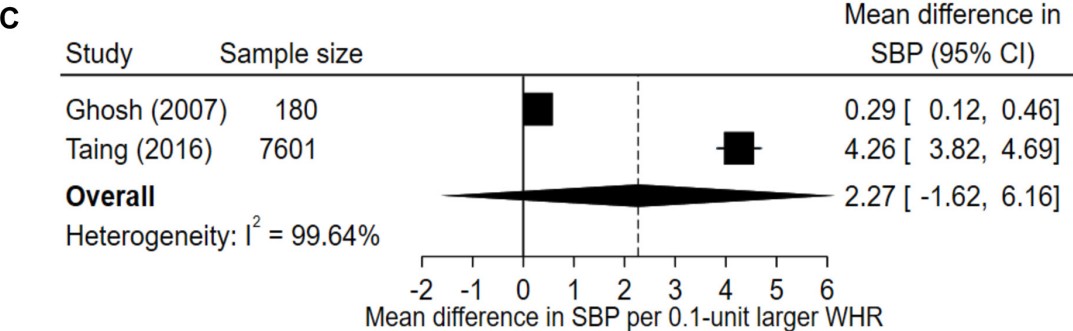

**Figure 2** Mean change in systolic blood pressure (SBP) per 5 kg/m² higher body mass index (BMI, A), 13 cm larger waist circumference (WC, B) and 0.1-unit larger waist-to-hip ratio (WHR, C). Random effects models were applied to four studies reporting associations of SBP and BMI, four studies reporting on SBP and WC, and two studies reporting on SBP and WHR. The total number of participants was 8143 for (A, B) and 7781 for (C).

was 1.65 (95% CI 1.55 to 1.75; figure 4A), despite three of the six studies making up this estimate concluding non-statistically significant findings.[43 45 46] Associations appeared weaker, but still statistically significant, for large versus normal WC (OR 1.48 (95% CI 1.21 to 1.80); figure 4B), and were not statistically significant for large vs normal WHR (OR 2.51 (95% CI 0.94 to 6.69); figure 4C). Heterogeneity (I²) was >75% in all models. Associations of BMI with CVD and of WC with CVD appeared stronger in fixed-effects models, and statistically significant for WHR (OR 1.50 (95% CI 1.43 to 1.57; online supplemental data S9).

### Sensitivity analyses
Sensitivity analyses were conducted by removing one study at a time and determining whether results were influenced by large studies or studies with extreme results. The results were not substantively different (online supplemental data S10).

### Study quality assessment and publication bias
Overall, the quality of the studies included in the review was average, with a mean score of 6.5/9 for cohort studies, 5.2/9 for case–control studies and 7/10 for cross-sectional studies (online supplemental data S4a/S4b/S4c). The domains in which studies lost points, depending on design, were principally sample representativeness,

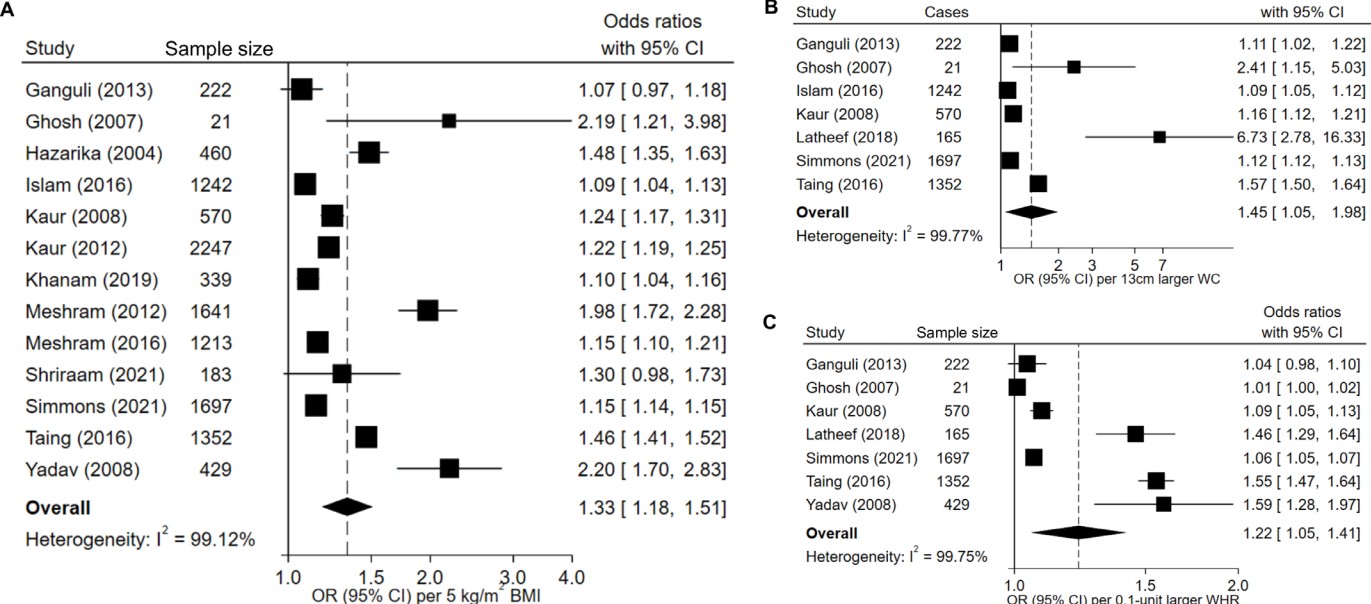

**Figure 3** Odds ratio (OR) of hypertension (HTN) per 5 kg/m² higher body mass index (BMI, A), 13 cm larger waist circumference (WC, B) and 0.1-unit larger waist-to-hip ratio (WHR, C). Random effects models were applied to 13 studies looking at BMI, seven studies looking at WC and seven studies looking at WHR. The total number of participants was 11 616 for (A), 4899 for (B) and 4456 for (C).

control for additional confounding factors, sample size and discussion of non-response rate.

Publication bias was assessed by inspecting funnel plots for asymmetry and using Egger's test. Funnel plots of studies reporting on risk of HTN associated with all three anthropometric indices (BMI, WC and WHR) showed significant skew to the right of the panel (figure 5). There were no studies in the lower left panel of the funnel, with most studies concentrating at the tip of the funnel and to the right of it. Funnel plots of studies reporting on the risk of CVD showed similar results (figure 5).

## DISCUSSION

The purpose of this review was to provide an overview of current literature examining the association of different anthropometric measures with CVD and HTN among South Asian populations. While BMI appeared marginally more strongly associated with CVD, WC appeared to be more strongly associated with higher SBP and DBP, as well as risk of HTN. Overall, there appears to be a limited amount of literature focusing on the shape of these associations, whereby studies only examined anthropometric measures as continuous variables (which assumes linearity) or as dichotomised variables, and an overall shape across the range of anthropometric measures was not assessed.

### Comparison of South Asian effects and other ethnicities

It is important to ascertain which anthropometric measures are better predictors of morbidity. Regarding CVD, the evidence is inconclusive, and varies depending on factors such as sex, ethnicity and subtype of CVD.[53] A large cross-sectional study, the International Day for the Evaluation of Abdominal Obesity study, looked at 168 000 participants across 63 countries and found WC to be a better predictor of CVD compared with BMI in men, but reported no significant difference between these measures in women (CVD ORs of BMI vs WC in men: 1.13 (95% CI 1.09 to 1.17) vs 1.24 (95% CI 1.19 to 1.28); CVD ORs of BMI vs WC in women: 1.20 (95% CI 1.16 to 1.24) vs 1.21 (95% CI 1.17 to 1.25)).[54] Specifically, for South Asians, the study concluded that the risk of CVD associated with a 1-SD increase in BMI was 1.26 (95% CI 1.17 to 1.35) for men and 1.26 (95% CI 1.18 to 1.35) for women, which was similar to 1.27 (95% CI 1.18 to 1.36) and 1.30 (95% CI 1.21 to 1.39), respectively, for WC.[54] A large prospective study of 0.5 million Chinese adults reported similar associations for stroke when comparing BMI and WC, whereas a large prospective study of 0.5 million adults in the UK reported that BMI was more strongly associated with myocardial infarction than WC in women, but with equivalent associations in men.[55 56]

There are several possible explanations for these results. Studies indicating stronger associations between CVD and WC, as opposed to BMI, support the theory that increased central obesity may be linked to systemic inflammation, a direct contributor to CVD risk. Additionally, central obesity is associated with higher levels of free fatty acids, which can interfere with insulin metabolism, leading to hyperinsulinaemia. This in turn promotes atherosclerosis, dyslipidaemia and the release of prothrombotic factors, which are linked to CVD. However, the results were inconsistent across region and sex regarding the relative importance of general versus

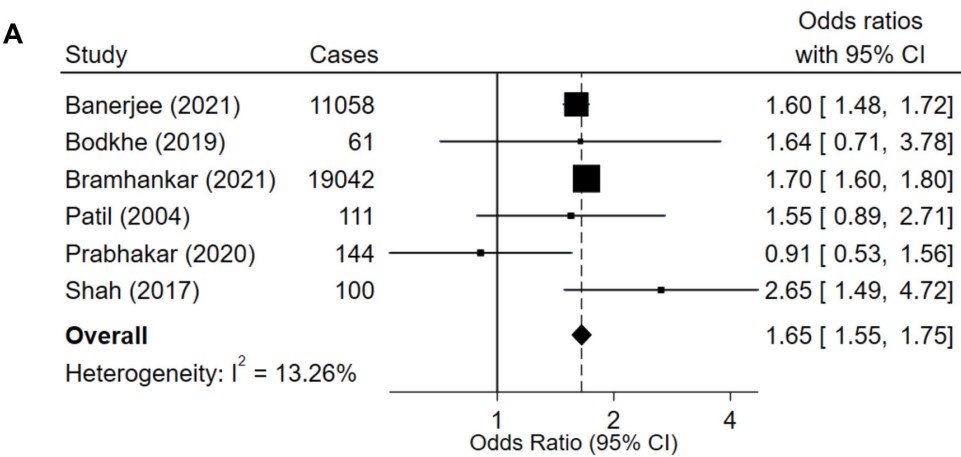

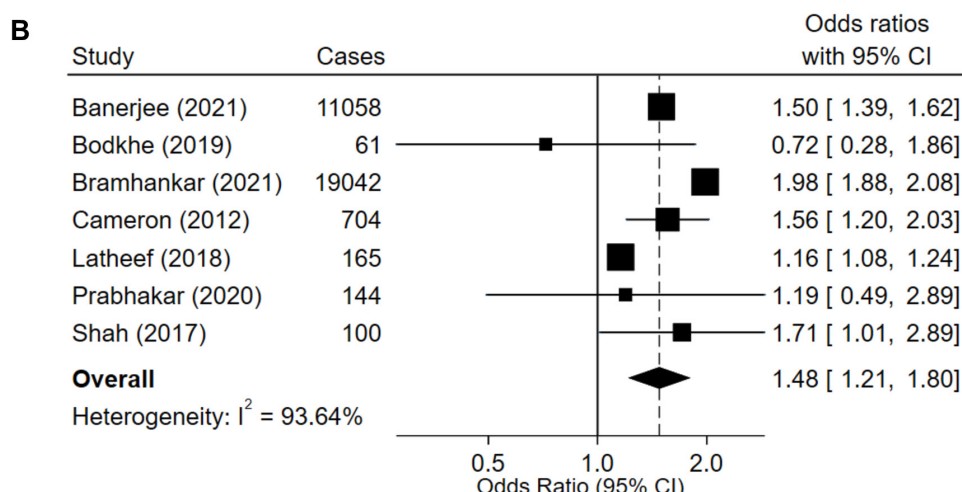

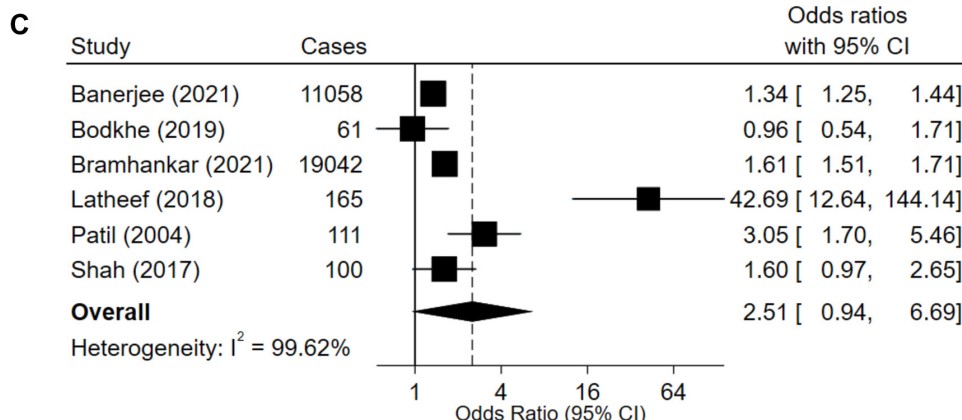

**Figure 4** Odds ratio (OR) of cardiovascular disease (CVD) for overweight (>25 kg/m²) versus normal (18.5–24.9 kg/m²) body mass index (BMI, A), large (≥80 cm in females, ≥90 cm in males) versus normal waist circumference (WC, B) and large (≥0.8 unit in females, ≥1.0 unit in males) versus normal waist-to-hip ratio (WHR, C). Random effects models were applied to six studies looking at BMI, seven studies looking at WC and six studies looking at WHR.

central adiposity for the risk of CVD, and it may be that subgroup-specific mechanisms and their relationships to CVD subtypes have not been fully elucidated yet.

In our study, the pooled estimates indicated that BMI, rather than WC, was more strongly associated with risk of CVD. The modestly improved prognostic value of BMI in South Asians may reflect the combined effects of height, fat mass and lean muscle mass that are each

individually associated with cardiometabolic risk in this ethnic group, but not represented by WC measures.[57] Recent large-scale research comparing Asian ethnic groups identified lean mass to be positively associated with SBP, triglycerides and haemoglobin A1C (HbA1c), which lie on the causal pathway of CVD.[57] Specifically, among Malay and Indian women, the associations of SBP, triglycerides and HbA1c with appendicular lean mass

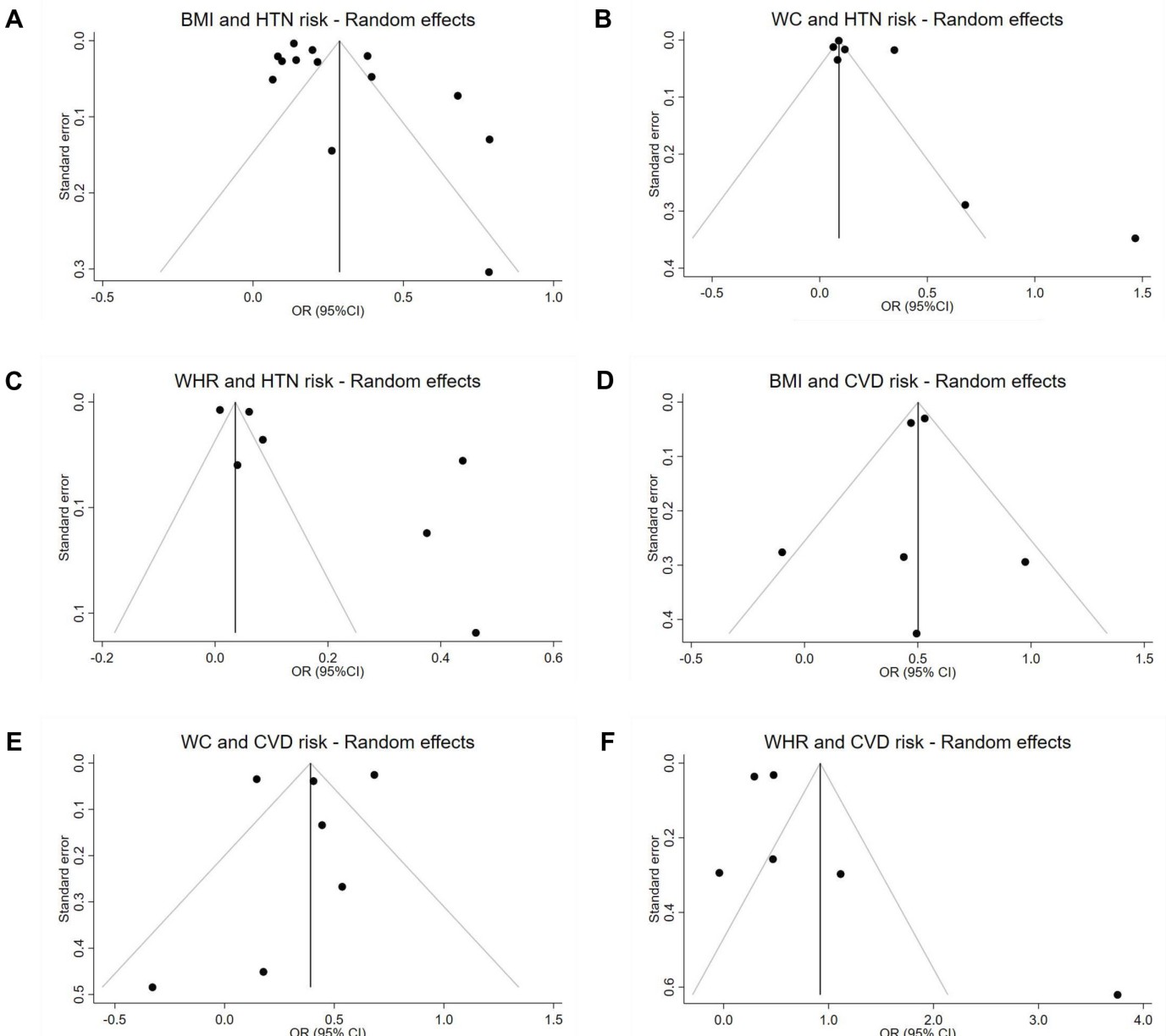

**Figure 5** Funnel plots of studies on BMI-HTN (A), WC-HTN (B), WHR-HTN (C) and BMI-CVD (D), WC-CVD (E) and WHR-CVD (F). Egger's test was b=2.27 (p=0.0025) for BMI and HTN, b=2.96 (p<0.001) for WC and HTN, and b=4.58 (p=0.003) for WHR and HTN. With regards to the CVD measures, Egger's test was b=0.43 (p=0.659) for BMI and CVD, b=0.42 (p=0.614) for WC and CVD and b=4.32 (p=0.0031) for WHR and CVD. Abbreviations: BMI, body mass index kg/m²; CVD, cardiovascular disease; HTN, hypertension; WC, waist circumference; WHR, waist-to-hip ratio.

were stronger than associations with BMI or fat mass. This suggests that future research needs to further unravel the correlates of BMI and WC in South Asians as it appears there are ethnic-specific mechanisms related to the risk of CVD. However, the results in this review comparing BMI to WC are still cautious as many of the included studies reported non-significant findings with CVD, likely due to low study power.[45 52] There is a need for larger prospective studies to directly compare anthropometric measures with measures of specific body composition (such as regional fat and lean mass depots) among South Asian populations (and their subtypes) to better understand the ethnic-specific aetiology of the risk of CVD.[58]

With regard to blood pressure, the wider literature concluded that BMI was positively associated with SBP (about 1 mmHg per 1 kg/m²),[8] which is slightly stronger than the results of the present study where a mean difference of about 3 mmHg in SBP per 5 kg/m² higher BMI was found. A large study on 0.5 million Chinese adults also concluded that BMI was more strongly associated with higher SBP than both WC and WHR.[59] However, the present study concluded that WC showed much stronger associations with blood pressure than both BMI and WHR. This is likely because South Asians, compared with other ethnicities, may manifest disproportionately larger WC for equivalent

BMIs due to a greater propensity to store visceral fat.[60] Further research has concluded that fat distribution, specifically central adipose tissue, may impact blood pressure variability over short-term and long-term periods, with greater amounts of visceral fat linked to elevated but less variable blood pressure, and thus to incidence of HTN.[61] From an epidemiological standpoint, this could explain why different associations have been observed in different ethnicities. Several studies also concluded that South Asians appear to have higher lipid and insulin levels compared with Europeans of the same WC and WHR.[62–64] Asian Indians tend to have greater visceral and total body fat, which is less evident from BMI measurements and differs from the typical Western build.[65] In turn, increased visceral fat can cause insulin resistance, dyslipidaemia and inflammation, which may lead to metabolic disorders such as HTN. The present study concluded that WC was more strongly associated with risk of HTN than BMI (OR WC 1.45 (95% CI 1.05 to 1.98); OR BMI 1.33 (95% CI 1.18 to 1.51)), which is in keeping with this theory. While BMI has long been used as a general indicator of obesity, the recognition that WC may be more strongly associated with HTN, and that HTN is linked to increased morbidity, may allow for more accurate risk stratification and preventative interventions to address the burden of downstream CVD risk. This is particularly relevant to South Asian populations for whom previous research has concluded that blood pressure is strongly and positively associated with CVD mortality, but that BMI is little related to CVD mortality, despite higher BMI being a strong determinant of higher blood pressure and consequently HTN.[8]

The INTERHEART case-control study, which compared populations, concluded that in all subgroups, but particularly among South Asians and mixed-race Africans, WHR was a better predictor of CVD than BMI.[66] In the present study, however, WHR was generally weaker than BMI and WC in associations with SBP, CVD and HTN, and across fixed-effects models. In INTERHEART, both WC and WHR were strongly associated with the risk of MI, but unlike BMI, this relationship was unaffected by mutual adjustment, suggesting there is a degree of independence between measures of adiposity in predicting the risk of myocardial infarction, stressing the relative importance of central adiposity measures.[66] Thus, because SBP is strongly associated with CVD risk, some adverse or protective correlate of low BMI is likely associated with CVD, particularly among South Asians.[67 68]

Finally, the shape of the associations of different anthropometric indices and CVD among South Asian populations has been scarcely analysed across the literature, with most studies only examining anthropometric measures as dichotomous variables or by calculating risks for continuous measures that assume linearity. It was, therefore, not possible to review results across the range in comparison

to the wider literature. Given the generally flat associations concluded by the Chennai Prospective Study, a comparison of shape across the range would make for a useful analysis.

### Study quality assessment
The quality of the studies included in this review was average, with most studies scoring five or six out of nine or ten possible total points on the Newcastle-Ottawa scale. Several studies recorded CVD outcomes based on self-report or verbal autopsy, largely due to local lack of national registries or reported death certification. Eight studies investigating CVD endpoints such as stroke, myocardial infarction or death from CVD were case-control or cross-sectional in design, and therefore, potentially limited by reverse causality. While the included studies are limited in some respects, the populations within each study were homogenous in terms of age, sex ratio and education, indicating potential internal validity.

### Strengths and limitations of this review
This meta-analysis has several strengths. The overall large size enables assessment of the relationship between different anthropometric measures and CVD, with an appreciation of all relevant literature on this topic. Despite significant heterogeneity, sensitivity analyses excluding large studies contributing the most to the models showed that associations remained largely unchanged.

Due to the nature of the studies and the extent of statistical heterogeneity observed, this systematic review also has limitations. First, while the review aims to assess risk across South Asia, most studies were conducted in India, thus limiting generalisability of findings to the rest of the subcontinent. Additionally, funnel plots of studies reporting on risk of HTN or CVD associated with all three anthropometric indices (BMI, WC and WHR) showed possible publication bias (figure 5). This may have impacted the ability to accurately synthesise the direction and strength of associations. Nevertheless, this reflects the current body of evidence, and highlights an important gap in the literature. Second, it may have been beneficial to exclude cross-sectional and case-control studies in examining the association of anthropometric measures with CVD endpoints such as MI and CVD to minimise reverse causality. However, a large proportion of the identified evidence was based on these study designs and excluding such studies would have skewed findings, thus biasing results. Third, despite ensuring the most fully adjusted models were used for the meta-analyses, we cannot rule out the possibility that observed associations are confounded by unmeasured factors such as physical activity and diet, including dietary salt consumption. Few studies adjusted for physical activity—a known confounder of cardiovascular health. The majority of studies also failed to control for other important confounders, for example, menopause status among females or dietary salt consumption, which are related independently to both anthropometric indices and CVD.[26 51]

## CONCLUSION

From a clinical point of view, health practitioners should be made aware of ethnic variations in CVD risk and how these relate to different measures of anthropometry. There is scope for measures such as WC and WHR to become routinely included in health records, alongside BMI, if these are truly deemed stronger CVD predictors. However, these measures would need to be robustly measured, which is not always straightforward in busy clinical environments and one of the reasons why BMI is more widely employed. Development of a point-of-care CVD risk score based on these measures may also prove an effective population-level prevention strategy.

Ultimately, large prospective studies among South Asian populations are required to clarify whether measures of central adiposity may be better predictors of CVD. Ideally, these studies would directly compare different measures of adiposity with risk over time. There is also potential value in imaging-based studies to characterise the distribution of adipose tissue more reliably. More considered cut-offs of different body composition measures, which consider location of fat deposition, may be needed, as well as an assessment of the shape of the relationship across the full range. Given the high prevalence of CVD globally, and the rapidly increasing prevalence among South Asian populations, this may have important implications from a public health perspective with potential to achieve better-targeted CVD primary prevention.

**Contributors** FR, JC and FB contributed to the design of the study and the statistical analysis plan. FR and ASO conducted the analysis. All authors (FR, ASO, BB, FB and JC) contributed to the interpretation of the analysis and the presentation of results. FR was responsible for drafting the manuscript. All authors contributed to reviewing and editing the manuscript, and all authors have agreed to the final version of the manuscript. FR and JC accept responsibility as the guarantors of this study.

**Funding** The study was funded by core support from the UK Medical Research Council (MRC), British Heart Foundation and Cancer Research UK to the Clinical Trial Service Unit and the MRC Population Health Research Unit, both now in the Nuffield Department of Population Health, University of Oxford (Oxford, UK).

**Competing interests** None declared.

**Patient and public involvement** Patients and/or the public were not involved in the design, or conduct, or reporting, or dissemination plans of this research.

**Patient consent for publication** Not applicable.

**Provenance and peer review** Not commissioned; externally peer reviewed.

**Data availability statement** Individual data should be requested from the original or parent study investigators of the studies included in this review.

**ORCID iDs**
Federica Re http://orcid.org/0000-0003-2264-9428
Ayodipupo S Oguntade http://orcid.org/0000-0001-8802-8590
Fiona Bragg http://orcid.org/0000-0002-9181-6886
Jennifer L Carter http://orcid.org/0000-0002-5298-4844

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
