## [Reviewer comments · BMJ Open]

ARTICLE DETAILS

TITLE (PROVISIONAL)	Associations of General and Central Adiposity with Hypertension and Cardiovascular Disease Among South Asian Populations: A Systematic Review and Meta-Analysis
AUTHORS	Re, Federica; Oguntade, Ayodipupo; Bohrmann, Bastian; Bragg, Fiona; Carter, Jennifer

VERSION 1 – REVIEW

REVIEWER	Herman, Saori Wendy Donald and Barbara Zucker School of Medicine at Hofstra/Northwell
REVIEW RETURNED	02-May-2023

GENERAL COMMENTS	No comments at this time.
---------------------------

REVIEWER	Mallinson, Poppy London School of Hygiene and Tropical Medicine Faculty of Epidemiology and Population Health, Department of Non-communicable Disease Epidemiology
REVIEW RETURNED	08-Jun-2023

GENERAL COMMENTS	This review addresses an interesting question and makes an useful addition to the literature on ethnic differences in the relationship between body composition and cardiovascular risk. Overall, it is well written and well conducted. However a few elements may need to be addressed before it can be published, mostly to do with interpretation and discussion of the results. 1. The study objective is to compare how strongly general (BMI) and central (WC/WHR) adiposity are associated with hypertension and CVD in South Asia. Accordingly, the main results presented are comparisons of the associations between CVD outcomes and BMI vs WC vs WHR, on continuous scales. However, to what extent are the intervals used to report effect size for each exposure (5 kg/m², 10cm and 0.1 units, respectively) directly comparable to each other? This will strongly determine the size of the association of each with outcomes. Comparability of these needs to be clearly justified, or else an alternative approach adopted. This issue also needs to be addressed when interpreting results and their limitations. (The rationale that these intervals were used for comparability with external studies does not follow here). 2. The discussion compares the results of the BMI associations in this study with other major studies, noting interesting discrepancies. However, I don't think these comparisons are valid or useful, as this study was specifically designed to compare different adiposity measures to each other. As a result, many
---

	studies in South Asian populations which only include one adiposity measure (e.g. BMI but not others) were excluded. These studies would need to be included if you were to make a meaningful comparison with the overall literature on the BMI-CVD association between populations. This would be a different research question, not well addressed by the design of this review. These comparisons and discussions of their possible explanations can be removed from the discussion. Instead, the discussion should focus on the results of the question being addressed by the review, for which in-depth discussion is currently lacking. For example, is there some suggestion that WC is particularly strongly associated with BP, but for other outcomes all anthropometric measures perform similarly? – and if so why might this be? Also, are similar patterns observed in non-South Asian ethnic groups (what are the relative sizes of these associations (BMI vs WC vs WHR) in non-South Asians)? 3. It is interesting to include hypertension alongside cardiovascular disease in the review, although could be expanded in the discussion how the results on these different conditions should be interpreted alongside each other, i.e. how these results might complement each other to aid understanding. And if relevant any specific considerations for relating to South Asian populations here. 4. From the funnel plots, it looks like publication bias may be present for some of the comparisons. This needs to be fully addressed in the discussion and limitations. 5. Minor comments on abstract: a. Sentence ending in “evidence on shape” not clear – shape of what? b. For results reported in abstract, number of studies contributing to each pooled effect can be mentioned.
--	---

REVIEWER	Li, Ruohong Microsoft Corp
REVIEW RETURNED	01-Aug-2023

GENERAL COMMENTS	This paper provides an overview of the literature on the association of different anthropometric measures (BMI, WC, and WHR) with CVD risk (SBP, DBP, Hypertension, and CVD) among South Asians. The result shows both general and central adiposity had similar association with CVD, further study is required to clarify which measure is more informative. The authors provided a comprehensive search strategy and inclusion/exclusion criteria. My only comments to the authors are: 1. In the third paragraph on page 7, please explain why the criterion is at least 50 events and 100 participants. 2. On Page 8, Data synthesis and analysis, it will be clearer to write down the random/fixed effect model formulas.
--

VERSION 1 – AUTHOR RESPONSE

Reviewer #1 - Prof. Saori Wendy Herman, Donald and Barbara Zucker School of Medicine at Hofstra/Northwell

No comments at this time.

Reviewer #2 - Ms. Poppy Mallinson, London School of Hygiene and Tropical Medicine Faculty of Epidemiology and Population Health

This review addresses an interesting question and makes an useful addition to the literature on ethnic differences in the relationship between body composition and cardiovascular risk. Overall, it is well written and well conducted. However a few elements may need to be addressed before it can be published, mostly to do with interpretation and discussion of the results.

1. The study objective is to compare how strongly general (BMI) and central (WC/WHR) adiposity are associated with hypertension and CVD in South Asia. Accordingly, the main results presented are comparisons of the associations between CVD outcomes and BMI vs WC vs WHR, on continuous scales. However, to what extent are the intervals used to report effect size for each exposure (5 kg/m², 10cm and 0.1 units, respectively) directly comparable to each other? This will strongly determine the size of the association of each with outcomes. Comparability of these needs to be clearly justified, or else an alternative approach adopted. This issue also needs to be addressed when interpreting results and their limitations. (The rationale that these intervals were used for comparability with external studies does not follow here).

Response 1: DONE – Measures of adiposity were included in the models as continuous variables to give the change in blood pressure / odds of HTN / odds of CVD per unit change in adiposity measure. To address the potential for limited comparability between these measures, we ensured that the measures of adiposity were all scaled to the same SD unit change. Scaling factors were based on the mean and SD values reported by Taing et al., 2016 (available at: DOI 10.1186/s12872-016-0424-y), since this study had the largest sample size (n=7601) and was also one of the only studies in the meta-analysis to include all three adiposity measures (BMI, WC, WHR). The extracted mean (SD) values for all three anthropometry measures reported by Taing et al., 2016 are displayed in the table below for reference. Explanation of this updated methodology is available in the Methods section of the main text (p.8, lines 13-21). Figures 2B, 3B, S6A/B/C, and S7B were also updated to reflect this.

	Sample size	BMI	WC	WHR
Male	3756	21.6 (4.2)	75.9 (12.2)	0.88 (0.081)
Female	3855	22.0 (4.9)	68.9 (12.0)	0.78 (0.076)
Pooled	7601	21.8 (4.6)	72.3 (12.1)	0.83 (0.079)

Scaling was rounded to the nearest whole number for ease of interpretation and understanding to the reader.

- For BMI, the pooled SD was 4.6kg/m², so a 5kg/m² change represents a 1.087 SD change. We felt it was important to keep BMI scaled to a 5 kg/m² change to allow comparisons between the results in this meta-analysis with previous research on BMI and CVD (References). We then scaled WC and WHR to the same SD unit change as BMI= 5 kg/m².
- For WC, the pooled SD was 12.1cm, so 12.1 x 1.087 = 13.15cm (rounded to 13cm change in WC).
- For WHR, the pooled SD was 0.079 units, so 0.079 x 1.087 = 0.09 units (rounded to 1 unit change in WHR).

Updated explanation in main text: “BMI was presented as a 5 kg/m² change to allow comparability with other large-scale studies which have used the same units.[1,2] Associations with WC and WHR were compared to those of BMI by scaling the measures to the same SD unit change. Scaling factors were based on the mean and SD reported by Taing et al., 2016 since this study had the largest sample size (n=7,601) and included all three adiposity measures of interest.[3] The pooled BMI SD in the study by Taing et al., 2016 is 4.6 kg/m², so a change in 5 kg/m² represents a 1.087 SD change. Accordingly, central adiposity measures were then scaled to the same SD change as 5 kg/m² BMI, which corresponded to a 13 cm increase in WC and 0.1-unit change in WHR.” (p.8, lines 16-25).

2. The discussion compares the results of the BMI associations in this study with other major studies, noting interesting discrepancies. However, I don't think these comparisons are valid or useful, as this study was specifically designed to compare different adiposity measures to each other. As a result, many studies in South Asian populations which only include one adiposity measure (e.g. BMI but not others) were excluded. These studies would need to be included if you were to make a meaningful comparison with the overall literature on the BMI-CVD association between populations. This would be a different research question, not well addressed by the design of this review. These comparisons and discussions of their possible explanations can be removed from the discussion. Instead, the discussion should focus on the results of the question being addressed by the review, for which in-depth discussion is currently lacking. For example, is there some suggestion that WC is particularly strongly associated with BP, but for other outcomes all anthropometric measures perform similarly? – and if so, why might this be? Also, are similar patterns observed in non-South Asian ethnic groups (what are the relative sizes of these associations (BMI vs WC vs WHR) in non-South Asians)?

Response 2: DONE – We only included studies that had at least two measures of adiposity to ensure a more reliable comparison with standardized methods. Given that the literature in this area tends to be limited, that ethnic variation among South Asian populations is high, and that studies in these populations and on this topic tend to be relatively small, including studies with at least two measures of adiposity helps ensure strong within-study comparison of effect.

In line with the suggested edit, we removed the part of the discussion which compared results of this study to studies conducted in other ethnicities which only include one adiposity measure. We have replaced this with a more adequate comparison with studies that included more than one anthropometric measure of interest. We have also added a section addressing whether similar patterns are observed in non-South Asian ethnic groups.

This now reads: *“It is important to ascertain which anthropometric measures are better predictors of morbidity. Regarding CVD, the evidence is inconclusive, and varies depending on factors such as sex, ethnicity, and subtype of CVD.[4] A large cross-sectional study, the International Day for the Evaluation of Abdominal Obesity (IDEA) study, looked at 168,000 participants across 63 countries and found WC to be a better predictor of CVD compared to BMI in men, but reported no significant difference between these measures in women (CVD ORs of BMI versus WC in men: 1.13 [95%CI: 1.09-1.17] versus 1.24 [95%CI: 1.19-1.28]; CVD ORs of BMI versus WC in women: 1.20 [95%CI: 1.16-1.24] versus 1.21 [95%CI: 1.17-1.25]).[5] Specifically, for South Asians, the study concluded that the risk of CVD associated with a 1-SD increase in BMI was 1.26 (95%CI: 1.17-1.35) for men and 1.26 (95%CI: 1.18-1.35) for women, which was similar to 1.27 (95%CI: 1.18-1.36) and 1.30 (95%CI: 1.21-1.39), respectively, for WC.[5] A large prospective study of 0.5 million Chinese adults reported similar associations for stroke when comparing BMI and WC, whereas a large prospective study of 0.5 million adults in the United Kingdom reported that BMI was more strongly associated with myocardial infarction than WC in women, but with equivalent associations in men.[6,7]*

There are several possible explanations for these results. Studies indicating stronger associations between CVD and WC, as opposed to BMI, support the theory that increased central obesity may be linked to systemic inflammation, a direct contributor to CVD risk. Additionally, central obesity is associated with higher levels of free fatty acids, which can interfere with insulin metabolism, leading to hyperinsulinemia. This in turn promotes atherosclerosis, dyslipidaemia, and the release of pro-thrombotic factors, which are linked to CVD. However, the results were inconsistent across region and sex regarding the relative importance of general versus central adiposity for the risk of CVD, and it may be that subgroup-specific mechanisms and their relationships to CVD subtypes have not been fully elucidated yet.

In our study, the pooled estimates indicated that BMI, rather than WC, was more strongly associated with risk of CVD. The modestly improved prognostic value of BMI in South Asians may reflect the combined effects of height, fat mass, and lean muscle mass that are each individually associated with cardiometabolic risk in this ethnic group, but not represented by WC measures.[8] Recent large-scale research comparing Asian ethnic groups identified lean mass to be positively associated with SBP, triglycerides, and HbA1c, which lie on the causal pathway of CVD.[57] Specifically, among Malay and Indian women, the associations of SBP, triglycerides, and HbA1c with appendicular lean mass were stronger than associations with BMI or fat mass. This suggests that future research needs to further unravel the correlates of BMI and WC in South Asians as it appears there are ethnic-specific mechanisms related to the risk of CVD. However, the results in this review comparing BMI to WC are still cautious as many of the included studies reported non-significant findings with CVD, likely due to low study power.[9, 10] There is a need for larger prospective studies to directly compare anthropometric measures with measures of specific body composition (such as regional fat and lean mass depots) among South Asian populations (and their subtypes) to better understand the ethnic-specific aetiology of the risk of CVD.[11] (p.13-14, lines 5-30 and 1-20, respectively).

3. It is interesting to include hypertension alongside cardiovascular disease in the review, although could be expanded in the discussion how the results on these different conditions should be interpreted alongside each other, i.e. how these results might complement each other to aid understanding. And if relevant any specific considerations for relating to South Asian populations here.

Response 3: DONE – we have included a more comprehensive explanation for how results of the hypertension component may complement CVD findings. Importantly, our study found that BMI was more strongly associated with risk of CVD than WC, but that this relationship was reversed for HTN (i.e., WC was more strongly associated with risk of HTN than BMI), even though HTN is often a precursor for CVD. The Chennai Prospective study from India concluded similar findings, specifically that blood pressure was strongly and positively associated with CVD mortality, but that BMI was little related to CVD mortality, despite higher BMI being a strong determinant of elevated blood pressure and consequently HTN. Notably, the Chennai Prospective study did not compare measures of central adiposity, so our study extends these findings. Because BMI does not distinguish where fat and lean mass are stored, it may fail to account for the increased risk of abdominal fat. Ultimately, the fact that WC was found to be more strongly associated with HTN than BMI may also have implications for early intervention and prevention of downstream CVD events.

Updated explanation in main text:

- *“Further research has concluded that fat distribution, specifically central adipose tissue, may impact blood pressure variability over short-and long-term periods, with greater amounts of visceral fat linked to elevated but less variable blood pressure, and thus to incidence of hypertension.[12]* (p.14-15, lines 31-3)
- *“In turn, increased visceral fat can cause insulin resistance, dyslipidaemia, and inflammation, which may lead to metabolic disorders such as hypertension. The present study concluded that WC was more strongly associated with risk of HTN than BMI (OR WC: 1.45 [95%CI: 1.05-1.98]; OR BMI: 1.33 [95%CI: 1.18-1.51]), which is in keeping with this theory. While BMI has long been used as a general indicator of obesity, the recognition that WC may be more strongly associated with HTN, and that HTN is linked to increased morbidity, may allow for more accurate risk stratification and preventative interventions to address the burden of downstream CVD risk. This is particularly relevant to South Asian populations for whom previous research has concluded that blood pressure is strongly and positively associated*

with CVD mortality, but that BMI is little related to CVD mortality, despite higher BMI being a strong determinant of higher blood pressure and consequently HTN.[13] (p.15, lines 8-20).

4. From the funnel plots, it looks like publication bias may be present for some of the comparisons. This needs to be fully addressed in the discussion and limitations.

Response 4: DONE – we have addressed the issue of publication bias as a limitation in the Discussion section under the “strengths and limitations of this review” section. This now reads: *“Additionally, funnel plots of studies reporting on risk of HTN or CVD associated with all three anthropometric indices (BMI, WC, and WHR) showed possible publication bias (Figure 5). This may have impacted the ability to accurately synthesize the direction and strength of associations.”* (p.17, lines 2-5).

Minor comments on abstract:

5a. Sentence ending in “evidence on shape” not clear – shape of what?

This statement was referring to evidence regarding the shape of the association of adiposity with CVD across the range of values of adiposity. However, as research was lacking on this topic and is not included in the results of the abstract, we have removed this statement from the abstract.

Response 5a: DONE – amended sentence in abstract reads: *Studies reported generally higher risks of hypertension and CVD at higher adiposity levels.* (p.2, lines 19-20).

5b. For results reported in abstract, number of studies contributing to each pooled effect can be mentioned.

Response 5b: PARTIALLY DONE – to avoid confusion for the reader given the already high propensity of data in the abstract, and to ensure that the word count fits within the 300-word limit set by BMJ Open, we have opted to not include the number of studies contributing to each pooled effect in the abstract. However, we do believe that this, along with the number of participants within each included study, is important information. The number of studies contributing to each pooled effect size is therefore reported in the legend of each figure, while the number of participants within each study is directly reported in the figures themselves (see p.18 for figure legends).

Reviewer #3 - Dr. Ruohong Li, Microsoft Corp

This paper provides an overview of the literature on the association of different anthropometric measures (BMI, WC, and WHR) with CVD risk (SBP, DBP, Hypertension, and CVD) among South Asians. The result shows both general and central adiposity had similar association with CVD, further study is required to clarify which measure is more informative. The authors provided a comprehensive search strategy and Inclusion/exclusion criteria. My only comments to the authors are:

1. In the third paragraph on page 7, please explain why the criterion is at least 50 events and 100 participants.

Response 1: DONE – Small studies are more likely to find spurious associations just by chance, and those with statistically significant or strong associations are more likely to get published. We set a criterion for a minimal sample size required for included studies as we wanted to avoid potential publication bias whereby small studies with larger-than-average effect sizes were included, hence skewing our pooled estimates in the meta-analysis. The threshold we chose was arbitrary, but we sought to balance the desire of minimizing publication bias with the inclusion of a comprehensive range of studies with valid results. Since the included studies of CVD used logistic regression, we were also cautious when including small studies as logit coefficients are typically biased

overestimates of true associations in small to moderate-sized data sets, where events per variable are less than 50[14], and many methods for meta-analysis are based on large-sample approximations that can give misleading results for rare events.[15]

Amended explanation in methods reads: “*To limit potential publication bias from the inclusion of small studies with chance findings that reported stronger-than-average results, studies examining clinical endpoints (e.g., CVD, hypertension) were included in the review only if they included at least 50 events, while those examining continuous outcomes (e.g., blood pressure) were included if they included at least 100 participants.*” (p.7, lines 11-16).

2. On Page 8, data synthesis and analysis, it will be clearer to write down the random/fixed effect model formulas.

Response 2: DONE – formulas for the random and fixed effects models have been included in the Data Supplement (S11. *Random and fixed effects model formulas*, p.23 of the Data Supplement). The main document now reads: “Formulas for random and fixed effects models are included in the Data Supplement (Data Supplement S10)”.

REFERENCES

1. Chen Y, Copeland WK, Vedanthan R, Grant E, Lee JE, Gu D, et al. Association between body mass index and cardiovascular disease mortality in east Asians and south Asians: pooled analysis of prospective data from the Asia Cohort Consortium. *BMJ*. 2013;347.
2. Di Angelantonio E, Bhupathiraju SN, Wormser D, Gao P, Kaptoge S, de Gonzalez AB, et al. Body-mass index and all-cause mortality: individual-participant-data meta-analysis of 239 prospective studies in four continents. *The Lancet*. 2016;388:776–86.
3. Taing K, Farkouh M, Moineddin R, Tu J, Jha P. Age and sex-specific associations of anthropometric measures of adiposity with blood pressure and hypertension in India: a cross-sectional study. *BMC Cardiovasc Disord*. 2016;16.
4. Goh LGH, Dhaliwal SS, Welborn TA, Lee AH, Della PR. Anthropometric measurements of general and central obesity and the prediction of cardiovascular disease risk in women: a cross-sectional study. *BMJ Open* [Internet]. 2014 [cited 2021 May 4];4. Available from: <https://bmjopen.bmj.com/lookup/doi/10.1136/bmjopen-2013-004138>
5. Balkau B, Deanfield J, Després J, Bassand J, Fox K, Smith S, et al. International Day for the Evaluation of Abdominal Obesity (IDEA): A study of waist circumference, cardiovascular disease, and diabetes mellitus in 168,000 primary care patients in 63 countries. *Circulation*. 2007;116:1942–51.
6. Chen Z, Iona A, Parish S, Chen Y, Guo Y, Bragg F, et al. Adiposity and risk of ischaemic and haemorrhagic stroke in 0.5 million Chinese men and women: a prospective cohort study. *The Lancet Global Health*. 2018;6:e630–40.
7. Peters S, Bots S, Woodward M. Sex Differences in the Association Between Measures of General and Central Adiposity and the Risk of Myocardial Infarction: Results From the UK Biobank. *JAHA* [Internet]. 2018 [cited 2023 Oct 27];7. Available from: <https://www.ahajournals.org/doi/10.1161/JAHA.117.008507>
8. Wells J. Commentary: The paradox of body mass index in obesity assessment: not a good index of adiposity, but not a bad index of cardio-metabolic risk. *Int J Epidemiol*. 2014;43:672–4.
9. Nishtar S, Wierzbicki A, Lumb P, Lambert-Hamill M, Turner C, Crook M, et al. Waist-hip ratio and low HDL predict the risk of coronary artery disease in Pakistanis. *Current Medical Research and Opinion*. 2008;20:55–62.
10. Patil S, Joshi R, Gupta G, Reddy M, Pai M, Kalantri S. Risk factors for acute myocardial infarction in a rural population of central India: A hospital-based case–control study. *Nat Med J India*. 2004;17.

11. Li X, Qi L. Gene–Environment Interactions on Body Fat Distribution. *IJMS*. 2019;20:3690.
12. Levelt E, Pavlides M, Banerjee R, Mahmood M, Kelly C, Sellwood J, et al. Ectopic and Visceral Fat Deposition in Lean and Obese Patients With Type 2 Diabetes. *J Am Coll Cardiol*. 2016;68:53–63.
13. Gajalakshmi V, Lacey B, Kanimozhi V, Sherliker P, Peto R, Lewington S. Body-mass index, blood pressure, and cause-specific mortality in India: a prospective cohort study of 500 810 adults. *Lancet Glob Health*. 2018;6:e787–94.
14. Van Smeden M, De Groot J, Moons K, Collins G, Altman D, Eijkemans M, et al. No rationale for 1 variable per 10 events criterion for binary logistic regression analysis. *BMC Med Res Methodol*. 2016;16:163.
15. Deeks J, Higgins J, Altman D. Chapter 10: Analysing data and undertaking meta-analyses [Internet]. 2023 [cited 2023 Oct 30]. Available from: <https://training.cochrane.org/handbook/current/chapter-10>